# Neurosyphilis Presenting as Syndrome of Limbic Encephalitis Mimicking Herpes Simplex Virus Neuro-Infection Diagnosed Using CXCL13 Point-of-Care Assay—Case Report

**DOI:** 10.3390/brainsci13030503

**Published:** 2023-03-16

**Authors:** Eliška Marešová, Stanislav Šutovský, Hana Štefucová, Alena Koščálová, Peter Sabaka

**Affiliations:** 1Department of Infectology and Geographical Medicine, Faculty of Medicine, Comenius University in Bratislava, 831 01 Bratislava, Slovakia; elikrajcovicova@gmail.com (E.M.);; 21st Department of Neurology, Faculty of Medicine, Comenius University in Bratislava, 811 07 Bratislava, Slovakia

**Keywords:** limbic encephalitis, *Treponema pallidum*, syphilis, encephalitis, herpes simplex virus, CXCL-13

## Abstract

The syndrome of limbic encephalitis is a severe clinical condition with heterogenous aetiopathogenesis. A common pathogen causing the infectious syndrome of limbic encephalitis is herpes simplex virus (HSV), but rare cases caused by *Treponema pallidum* have also been reported. We present the case of a 46-year-old man who presented with sudden onset of headaches, nausea, vomiting, and short-term loss of consciousness with clonic convulsions and subsequent disorientation and aphasia. Examination of the cerebrospinal fluid (CSF) revealed lymphocytic pleocytosis and magnetic resonance of the brain revealed bilateral temporal lesions. Clinical, radiologic, and biochemical examinations of CSF suggested encephalitis caused by HSV. However, the positivity of CXCL-13 chemokine in the CSF by a rapid point-of-care assay suggested active spirochetal infection and led to further serologic investigation. The definitive diagnosis of neuro-syphilis was concluded by positive intrathecal synthesis of immunoglobulins against *Treponema pallidum*. Penicillin therapy led to a rapid improvement, and the patient was discharged home after three weeks. Due to memory problems and irritability, after eighteen months, he came for a follow-up neurological and psychological examination. The psychological examination revealed a significant deficit in executive functions and behavioural changes. Neurosyphilis should be considered in the differential diagnosis of limbic encephalitis with lymphocytic pleocytosis in cerebrospinal fluid, and CXCL-13 may help to achieve diagnosis.

## 1. Introduction

Syphilis is one of the most important sexually transmitted diseases, which despite effective and accessible antimicrobial treatment, still represents a global public health problem [1,2]. The incidence of syphilis in high-income countries decreased in the 1980s and 1990s due to changes in sexual behaviour associated with the emergence of the HIV pandemic, however, it has started to rise during the last two decades [3]. Each year, there are several million new infections reported worldwide [4]. In high-income countries, syphilis occurs at low rates in the general population, and a higher prevalence is only observed in specific populations (MSM, high-frequency sexual contact, and sex workers). In low-income countries, while the prevalence is higher in high-risk populations such as sex workers, it is also quite common in the general population [5]. Syphilis is a chronic infectious disease caused by spirochete *Treponema pallidum*. It is almost exclusively acquired sexually. Spirochetes access the body through the skin or mucous lesions. After proliferation in the site of inoculation and nearby lymphatic tissue, bacteria disseminate and may affect virtually all organs, causing obliterative endarteritis and necrotic changes in terminal arterioles. In later stages, localised inflammatory lesions might arise in affected organs [1,2]. The clinical presentation of syphilis is extremely diverse, and the symptoms might occur many months or even years after the initial infection. Therefore, it might be overlooked easily. Especially, late neurosyphilis might cause great differential diagnostic challenges because it often lacks specific signs or symptoms and can mimic a wide variety of other neurologic conditions [6,7]. A lack of awareness of possible neurosyphilis might result in delayed or missed diagnoses [8,9]. Neurosyphilis is one of the most severe forms of *Treponema pallidum* infection [6]. The typical neurological manifestations of neurosyphilis described in older textbooks (tabes dorsalis, progressive paralysis, and Argyll Robertson pupils) are now rarely seen in developed countries [1,2]. CNS involvement usually presents as single or multiple inflammatory lesions [10]. Neurosyphilis might rarely present as temporal lobe and limbic system involvement with characteristic radiologic traits of encephalitis caused by other infectious agents, especially herpes simplex virus (HSV) or autoimmune limbic encephalitis (LE). This may lead to an incorrect initial diagnosis, especially if neurosyphilis is not considered [11,12,13,14,15,16]. LE is a severe clinical condition characterized by inflammation of the limbic system. Symptoms of LE often involve seizures, behavioural changes, and retrograde amnesia. LE is usually autoimmune, but the spectrum of possible etiopathogenesis of LE syndrome is quite broad, including various infectious diseases [14,15]. HSV is the most common aethiopathogen of the infectious syndrome of LE. Because of a wide spectrum of possible aetiologies, a differential diagnosis of patients presenting with symptoms of LE and radiologic evidence of inflammation of the limbic system is often challenging [11]. The intrathecal synthesis of C-X-C motif ligand 13 (CXCL13) has been described in spirochetes infections such as neuroborreliosis and neurosyphilis and might be used as a complementary biomarker of spirochaetal neuro-infection [17]. Because of the high sensitivity and specificity for spirochetal infection of the CNS, the point-of-care tests for CXCL13 in CSF are regarded as a valuable tool for rapid diagnosis of neuroborreliosis [18,19,20]. We present a case of neurosyphilis mimicking the clinical syndrome of limbic system encephalitis caused by HSV, in which the positivity of CXCL13 in CSF led to the early administration of appropriate antimicrobial therapy and to the final diagnosis.

## 2. Case Presentation

A 46-year-old, previously healthy Caucasian male (ethnic group—Slovak, West Slavic region), presented to the Emergency department of the Department of Infectology and Geographical Medicine, University Hospital in Bratislava, with a history of sudden onset of headaches, nausea, vomiting, and collapsed, with subsequent disorientation, aphasia, and amnesia. According to the witnesses, the patient suddenly collapsed and lost consciousness with tonic and clonic convulsions. The seizures lasted about one minute. After regaining consciousness, the patient was disoriented and suffered from retrograde amnesia, mild motoric aphasia, and a headache. Objective examination revealed neck stiffness (resistance to flexion 2 cm of the sternum), otherwise, it was unremarkable. Body temperature and vital signs were unremarkable. After admission, the disorientation resolved, however, the patient’s behaviour possessed apparent prefrontal traits (tactlessness and facetiousness). These symptoms were consistent with the presentation of LE. A blood count examination revealed mild monocytosis (850 cells/mL, reference range: 100–700 cells/mL). The biochemical examination result including C-reactive protein and procalcitonin was unremarkable. Computed tomography of the brain (CT) revealed a hypodense area in the white matter of the left temporal lobe and a smaller hypodense area in the left frontal lobe. Subsequently, magnetic resonance imaging (MRI) of the brain revealed localized lesions with increased signal in T2 and fluid-attenuated inversion recovery (FLAIR) in the left temporal lobe extending to the insula and frontal lobe of 50 × 55 × 56 mm in diameter with the enhancement of the signal with gadolinium contrast. There was also a mild increase in the signal in T2 and FLAIR in the ventral parts of the right temporal lobe and cortically localised accumulation of hemosiderin in the left frontal lobe (Figure 1). The clinical findings were indicative of encephalitis affecting the limbic system. We performed a lumbar puncture, and an examination of the cerebrospinal fluid (CSF) revealed pleocytosis with 80 mononuclear per μL, 12 polymorphonuclears per μL, and 5 erythrocytes per μL. Total protein concentration was increased to 1 235 mg/L (reference range: 200–500 mg/L). The albumin in CSF level was increased to 441 mg/L (reference range: 120/300 mg/L) and the immunoglobulin G level was increased to 840 mg/L (reference range: 12–40 mg/L). The Immunoglobulin G index was 5.53. Reibers’ criteria for the intrathecal synthesis of immunoglobulins were positive [21]. Glycorrhachia and lactate concentrations in the CSF were unremarkable. The polymerase chain reaction was negative for HSV 1 and HSV 2 deoxynucleic acid in the CSF. We examined the patient’s CSF for CXCL13 chemokine by rapid point-of-care assay, ReaScan CXCL13 (Reagena Ltd., Toivala, Finland), which was highly positive by a semi-quantitative lateral flow essay. The CXCL13 assay is routinely used in our ward for patients presenting with serous meningitis and meningoencephalitis for the rapid diagnosis of spirochetal infection to quickly identify patients with probable Lyme disease. Because of the clinical presentation and MRI findings of lesions in the temporal lobes, which is the clinical presentation common in encephalitis caused by HSV, therapy by parenteral acyclovir was started. On the other hand, a high concentration of CXCL13 chemokine in the CSF suggested spirochete neuro-infection. We considered a diagnosis of neuroborreliosis, since the Borrelia burgdorferi is the common spirochetal pathogen causing neuro-infections in Central Europe [22]. Therefore, we also started a treatment of a dose of 2 g intravenous ceftriaxone every 24 h. Positivity of CXCL13 and clinical presentation that is atypical for neuroborreliosis also led to a further serologic examination for Treponema pallidum. The following day, a serological examination of the patient serum and CSF was negative for the intrathecal synthesis of immunoglobulins against HSV 1 and HSV 2 by electrochemiluminescence (ECLIA) Elecsys^®^ (Roche Diagnostics International Ltd., Rotkreuz, Switzerland). The serum titer for anti-HSV1/2 immunoglobulin G by ECLIA was 230 units/mL (cut off >22 units/mL), and the index for intrathecal synthesis was 0.9 (cut off >1.5). Therefore, we stopped acyclovir administration. Serological examination revealed positive non-treponemal rapid plasma reagin reaction (RPR) by the Mediace^®^ Syphilis Rapid Plasma Reagin automated tests (Sekisui Medical Co.; Tokyo, Japan), which was further confirmed by the positivity of a specific treponemal test, namely ECLIA Elecsys^®^ Syphilis (Roche Diagnostics International Ltd., Rotkreuz, Switzerland) and the Syphilis Immunoblot assay (Mikrogen Diagnostik, Neuried, Germany). The anti-treponema antibodies were detected in both the serum and CSF. The serum titer of anti-treponema antibodies by ECLIA was 152 units/mL (cut off >1.0 units/mL). The intrathecal synthesis of immunoglobulin G (IgG) against Treponema pallidum was highly positive, with an index exceeding 1:100 (cut off >1:2). We concluded the diagnosis of neurosyphilis and switched from ceftriaxone therapy to intravenous penicillin at the dose of 4 million units every 4 h. On day 4 of the treatment, the prefrontal symptoms and nuchal rigidity resolved. On day 13 of the treatment, the second MRI revealed no change in the hyper signal areas, but it confirmed the complete resolution of opacification of the cortex and meninges. After 14 days of intravenous penicillin, the patient was discharged. At the eighteen-month follow-up, the patient still suffered from frequent headaches, and a psychological examination revealed a mild amnestic deficit, increased hostility, irritability, and impulsivity corresponding with mild cognitive impairment of predominantly the frontal type accompanied by neuropsychiatric symptoms. MRI of the brain revealed the complete resolution of encephalitis.

## 3. Discussion

Neurosyphilis, especially at the late stages, might mimic various neurological syndromes and diseases and present a great diagnostic challenge. The symptoms are often non-specific and diverse. The recent review of 50 cases describes a wide variety of symptoms, which most common were seizures, limb weakness, headaches, vertigo, dysarthria, aphasia, and mental alterations [9]. Our case study patient presented with a headache, unconsciousness, seizures with subsequent disorientation, and prefrontal syndrome. The localized lesions in the brain caused by late neurosyphilis are usually visualized by MRI as ring-like, strip-like, or uniform lesions characterized by signal hyperintensity under T1-weighted images and hyperintensity under T2-weighted images, which are enhanced in gadolinium-enhanced T1-weighted images. CT usually reveals hypodense areas with possible calcifications or haemorrhages. The lesion is usually single, but there might be multiple lesions, and localisation is not uniform [10]. These changes, however, are not specific and might mimic other conditions. Neurosyphilis cases presenting as LE are rare. To our knowledge, there are three case reports of encephalitis affecting the limbic system caused by neurosyphilis that were initially misdiagnosed as HSV encephalitis [11,12,13]. There are also various cases of the syndrome of limbic encephalitis caused by neurosyphilis masquerading as autoimmune LE [14,15,16]. Our case and three previously published cases of *Treponema pallidum* encephalitis mimicking encephalitis caused by HSV featured key similarities. First, the MRI finding was characterized by a bilaterally increased signal in T2 and FLAIR MRI images in the temporal cortex, and unilateral or bilateral temporal involvement is typical for HSV encephalitis. Second, the most prominent clinical presentations were memory loss and confusion. Third, the CSF examination revealed pleocytosis with lymphocyte predominance and an elevated CSF protein concentration [11,12,13,14]. The originality of our case report is the utilisation of the CXCL13 assay to assess the intrathecal synthesis of CXCL13 chemokine at the time of presentation. The positivity of CXCL13 in the CSF suggested spirochaetal neuro-infection, and based on the presence of CXCL13 in the CSF, ceftriaxone therapy was started, and serologic testing for syphilis was performed. CXCL13, originally known as B-lymphocyte chemoattractant or B cell-attracting chemokine 13, is a chemokine secreted by stromal cells in B cell areas of secondary lymphoid tissues. CXCL13 strongly attracts B lymphocytes, promotes their migration to the secondary lymphoid tissue, and controls the organisation of B cells within lymphatic follicles. [17]. Its positivity is highly associated with a neuro-infection caused by Borrelia burgdorferi and Treponema pallidum. According to a metanalysis, its sensitivity and specificity for neuroborreliosis exceed 89%, 96%, respectively. Therefore, it is regarded as a useful adjunct in the diagnosis of acute neuroborreliosis. In spirochaetal neuro-infection, CXCL13 production has been identified in the lymphocyte aggregates in the CNS. It is presumed that it facilitates the migration and aggregation of B cells in the CNS [18]. On our ward, we implemented the point-of-care semiquantitative test for ReaScan CXCL13 for the rapid differential diagnosis of spirochetal neuro-infections. We routinely perform this assay for all patients with meningitis and meningoencephalitis with lymphocytic pleocytosis in CSF. The sensitivity of this assay in neuroborreliosis is 78%, and the specificity is 95%, and concordance with an enzyme-linked immunoassay is 87%. Therefore, it has been proposed as a valuable point-of-care test in patients with possible neuroborreliosis [19]. The Infectious Diseases Society of America (IDSA) guidelines for the diagnosis and treatment of Lyme disease recognizes the CXCL13 as a potential biomarker of acute neuroborreliosis. However, because of a lack of decisive evidence of its usefulness, it is not yet implemented in the diagnostic algorithm proposed by IDSA. Its most important shortcoming recognised by IDSA is the fact that CXCL13 in CSF is positive not just in Lyme disease, but also in other spirochetal neuro-infections such as neurosyphilis [20]. Therefore, the CXCL13 assay has been proposed as a valuable tool in the diagnosis of neurosyphilis. Its sensitivity in the diagnosis of neurosyphilis exceeds 85% and the specificity exceeds 89% in the setting of patients with possible Treponema pallidum infection and CNS involvement [23,24]. However, because it is positive in other spirochetal neuro-infections and other inflammatory diseases of CNS, the detection of anti-treponema antibodies is required to conclude the diagnosis. Indirect non-treponemal and treponemal tests to detect anti-treponemal antibodies are routinely used to diagnose *Treponema pallidum* infections. A positive screening, indirect, non-treponemal test should be followed by a confirmatory treponemal test, typically Treponema pallidum particle agglutination (TPPA) or ECLIA and Western blots. The positivity of the treponemal test represents the final diagnosis [1,2]. To date, there is insufficient evidence to exactly describe the role of the CXCL13 assay in the routine management of neurosyphilis, and CXCL13 examination is not recommended by the latest guidelines [25]. The advantage of the point-of-care assay for CXCL13 in CSF in patients with meningitis and meningoencephalitis is that it rapidly identifies the patients with a possible spirochetal neuro-infection. That may allow us to optimize further diagnostics and start the antimicrobial treatment early. However, the final diagnosis should be confirmed by serological testing. The case patient suffered from a long-term neurologic sequel characterized as mild cognitive impairment of predominantly frontal type accompanied by neuropsychiatric symptoms even 18 months after the hospital discharge. Behavioural changes and amnestic deficits are the common long-term sequelae of limbic encephalitis [26].

## 4. Conclusions

Neurosyphilis and its variable clinical image often mimic other diagnoses such as meningoencephalitis, epilepsy, ischemia, a haemorrhage, or intoxication. Treponema pallidum encephalitis located in the temporal lobes involving the limbic system often imitates clinical and radiologic traits of encephalitis caused by HSV or autoimmune LE. In these cases, the diagnosis of neurosyphilis should be considered. Examination of CXCL13 in CFS might help to establish the final diagnosis because it is highly associated with spirochaetal neuro-infections. The patient should always be followed up for possible neurologic or psychiatric sequelae. Residual cognitive and behavioural impairments may persist throughout life.

## Figures and Tables

**Figure 1 brainsci-13-00503-f001:**
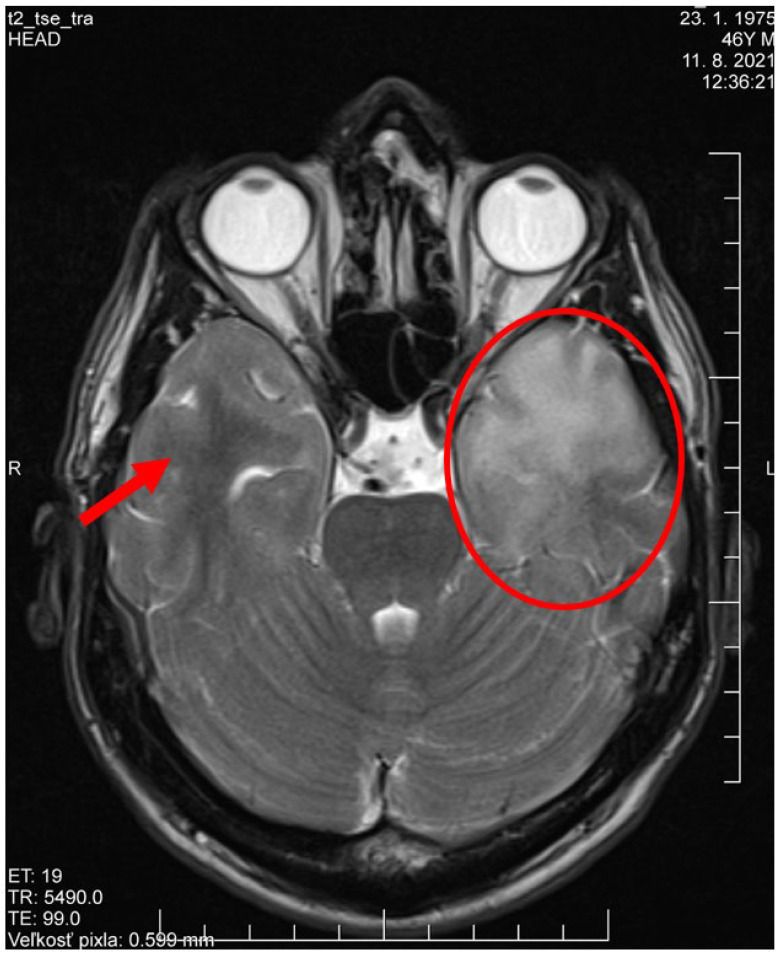
Magnetic resonance imagining of the brain (FLAIR and T2 weighted) at the time of presentation. It revealed localized lesions with increased signal in the left temporal lobe extending to the insula and frontal lobe of 50 × 55 × 56 mm in diameter with the enhancement of the signal with gadolinium contrast (circle). There is also a mild increase in the signal in the ventral parts of the right temporal lobe (arrow).

## Data Availability

The data presented in this study are available on request from the corresponding author. The data are not publicly available in order to protect the privacy of our patient.

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
