# Peer review of "Neurosyphilis Presenting as Syndrome of Limbic Encephalitis Mimicking Herpes Simplex Virus Neuro-Infection Diagnosed Using CXCL13 Point-of-Care Assay—Case Report"

_brainsci, 2023, doi:10.3390/brainsci13030503_

Round 1
Reviewer 1 Report
I have read with interest the case report and I have a couple of minor requests
1: I think that it could be useful to add information about the serology methods used, both for serum sample and CSF
2: The authors should comment about the importance of the involvement of CXCL-13 in neurosyphilis
Finally, I suggest to revise English, since there are minor errors
Author Response
Dear reviewer. We appreciate your valuable inputs
1: I think that it could be useful to add information about the serology methods used, both for serum sample and CSF
We added information about serologic methods used to analyze plasma and CSF covering particular serologic methods and manufacturers.
2: The authors should comment about the importance of the involvement of CXCL-13 in neurosyphilis
We commented on the potential of the CXCL13 assay in the management of neurosyphilis and Lyme disease in the discussion section.
Finally, I suggest to revise English, since there are minor errors
The English have been revised by a professional proofreader.
Reviewer 2 Report
Maresova et al report a case of neurosyphilis-associated autoimmune encephalitis initially suspected as herpes simplex encephalitis until availability of positive CXCL13-PCA results. They emphasize considering neurosyphilis as differential diagnosis of limbic encephalitis with lymphocytic CSF pleocytosis and the usefulness of CXCL13 as differential diagnostic biomarker.
The case report in its current form is suboptimal structured, partly redundant and the sequel of events confusing. Authors may want to focus on “the originality of our case report …”), improve rationale and sequence of decisions (f.e why did they test for CXCL13?), and remove parts which are not relevant to the sequel reaching the final diagnosis. The one-year follow-up visit accordingly only deserves short mention. The laboratory work-up leading to diagnosis on the other hand should be provided with details such as the test formats for intrathecal Treponema/HSV-specific antibody synthesis and results of the Reibergraphs such as QAlb and QIgG.
I definitely miss background information and discussion on the current state of the art, opportunities and limitations of using CXCL13 as supportive marker in the differential diagnosis of spirochetal infections. The point-of care assay they used and its validity should be explained.
Authors may want to reconsider their phrasing as diagnosing a condition as limbic encephalitis is different from a condition presenting as limbic encephalitis. As to their proposed key similarities between neurosyphilis and herpes viral encephalitis – MRI hyperintensities can be unilateral OR bilateral, not sole bilateral.
A figure illustrating the sequel would be helpful.
Minor Issues:
Please, remove information from the discussion which has been provided in the introduction.
Terms like “syphilitic” and “herpetic” are not too common, I suggest avoiding them.
CSF cell counts nowadays conventionally are specified as xy cells/µl not x/3 cells anymore.
There are several flaws in the text, authors should, please, check for those.
Author Response
Dear reviewer
We would like to thank you for your valuable input. We hope that we provided all necessary revisions. We belive that the revisions you suggested make the manuscript more valuable and eligible for publication in revised form.
The case report in its current form is suboptimal structured, partly redundant and the sequel of events confusing. Authors may want to focus on “the originality of our case report …”), improve rationale and sequence of decisions (f.e why did they test for CXCL13?), and remove parts which are not relevant to the sequel reaching the final diagnosis. The one-year follow-up visit accordingly only deserves short mention. The laboratory work-up leading to diagnosis on the other hand should be provided with details such as the test formats for intrathecal Treponema/HSV-specific antibody synthesis and results of the Reibergraphs such as QAlb and QIgG.
We deleted all redundant information from "Discussion". We made the report about the sequel much more bier. We revised the whole case report to be more informative and logic for the reader and deleted unimportant information. In the "Discussion", we commented on the originality of the study aimed at the potential role of CXCL13 assay in the management of neurosyphilis and neuro-infections in general. We provided the reader with the information regarding serologic methods used in a more detailed way (method used and manufacturer).
I definitely miss background information and discussion on the current state of the art, opportunities and limitations of using CXCL13 as supportive marker in the differential diagnosis of spirochetal infections. The point-of care assay they used and its validity should be explained.
We added information about the possible role of CXCL13 assay in the management of Lyme disease and neurosyphilis in the "Discussion". We also commented value and performance of the point-of-care assay we used.
Authors may want to reconsider their phrasing as diagnosing a condition as limbic encephalitis is different from a condition presenting as limbic encephalitis. As to their proposed key similarities between neurosyphilis and herpes viral encephalitis – MRI hyperintensities can be unilateral OR bilateral, not sole bilateral.
We rephrased the term "Limbic encephalitis" in the whole manuscript to distinguish between autoimmune limbic encephalitis and other clinical conditions presenting as "syndrome of limbic encephalitis" to avoid confusion. We also corrected the characteristics of neurosyphilis and HSV encephalitis to include also the syndrome with unilateral involvement.
A figure illustrating the sequel would be helpful.
Unfortunately, we don't have the footage from the psychological examination to demonstrate the sequel, so we are unable to provide it.
Minor Issues:
Please, remove information from the discussion which has been provided in the introduction.
We removed redundant information.
Terms like “syphilitic” and “herpetic” are not too common, I suggest avoiding them.
We removed these terms and used the correct terms (HSV encephalitis and Treponema pallidum encephalitis)
CSF cell counts nowadays conventionally are specified as xy cells/µl not x/3 cells anymore.
We provided the cell count in CSF in ul
There are several flaws in the text, authors should, please, check for those.
The text has been revised by a proofreader.
Round 2
Reviewer 2 Report
The manuscript improved – minor points remain:
Line 54/55: phrasing error
Line 58: phrasing error “the”
Line 99ff: CSF cell count specification are wrong – authors initially wrote f.e. 239/3 mononuclear per ml – cell counts are per µl not ml, and 239/3 ~ 80 cells/µl. The same is true for granulo (PMN) and erythrocytes. Please, correct that!!!
Line 100: Please provide normal ranges where pathological values are specified: i.e. blood monocytosis and lab CSF parameters (TP, albumin, IgG..)
Line 102: Authors may want to add the work about reibers criteria they referred to into their reference list
Line 113: Authors may want to specify “spirochete neuroinfection” why they started with tetracyclines since treponema are spirochetes too, but require different treatment.
Line 136: typo
Lines 177-181: the information about CXCL13 lacks the link to neuroinflammation. Lymph nodes are not in the brain…
Author Response
Dear reviewer
We greatly appreciate your valuable suggestions and revisions.
Line 54/55: phrasing error
- We corrected the phrasing error: CNS involvement usually presents as single or multiple inflammatory lesions.
Line 58: phrasing error “the”
- we deleted the the in the sentence.
Line 99ff: CSF cell count specification are wrong – authors initially wrote f.e. 239/3 mononuclear per ml – cell counts are per µl not ml, and 239/3 ~ 80 cells/µl. The same is true for granulo (PMN) and erythrocytes. Please, correct that!!!
- We corrected the values. The incorrect values were provided by an external laboratory and were incorrectly provided by mistake.
Line 100: Please provide normal ranges where pathological values are specified: i.e. blood monocytosis and lab CSF parameters (TP, albumin, IgG..)
- We provided the reference values.
Line 102: Authors may want to add the work about reibers criteria they referred to into their reference list
- We cited the paper describing Reiber criteria: Auer M, Hegen H, Zeileis A, Deisenhammer F. Quantitation of intrathecal immunoglobulin synthesis - a new empirical formula. Eur J Neurol. 2016;23(4):713-721. doi:10.1111/ene.12924
Line 113: Authors may want to specify “spirochete neuroinfection” why they started with tetracyclines since treponema are spirochetes too, but require different treatment.
- We specified that Lyme disease was suggested since Borrelia burgdorferi is the most common spirochaetal pathogen causing neuro-infection in our region.
Line 136: typo
- We corrected the typo
Lines 177-181: the information about CXCL13 lacks the link to neuroinflammation. Lymph nodes are not in the brain…
- We added: In spirochaetal neuro-infection, CXCL13 production has been identified in the lymphocyte aggregates in CNS. It is presumed that it facilitates the migration and aggregation of B-cells in the CNS.
We hope that the revised manuscript will meat the standards for case reports in Brain Sciences.